# The Degradation of Deoxynivalenol by Using Electrochemical Oxidation with Graphite Electrodes and the Toxicity Assessment of Degradation Products

**DOI:** 10.3390/toxins11080478

**Published:** 2019-08-19

**Authors:** Suli Xiong, Xiao Li, Changsong Zhao, Jingqi Gao, Wenjuan Yuan, Jie Zhang

**Affiliations:** Key Laboratory of Biological Resources and Ecological Environment of the Ministry of Education, College of Life Sciences, Sichuan University, Chengdu 610065, China

**Keywords:** deoxynivalenol, electrochemical oxidation degradation, graphite electrode, cell toxicity, wet distiller’s grain with solubles (WDGS)

## Abstract

Deoxynivalenol (DON) is a common mycotoxin, which is known to be extremely harmful to human and livestock health. In this study, DON was degraded by electrochemical oxidation (ECO) using a graphite electrode and NaCl as the supporting electrolyte. The graphite electrode is advantageous due to its electrocatalytic activity, reusability, and security. The degradation process can be expressed by first-order kinetics. Approximately 86.4% of DON can be degraded within 30 min at a potential of 0.5 V. The degradation rate reached 93.2% within 30 min, when 0.5 V potential was used for electrocatalyzing a 10 mg/L DON solution. The degradation rate of DON in contaminated wet distiller’s grain with solubles (WDGS) was 86.37% in 60 min. Moreover, results from the cell counting kit-8 (CCK-8) and 4,6-diamidino-2-phenylindole dihydrochloride (DAPI) staining assay indicated that ECO reduced the DON-induced cytotoxicity and apoptotic bodies in a gastric epithelial cell line (GES-1) compared to the DON-treated group. These findings provide new insights into the application of ECO techniques for degrading mycotoxins, preventing food contamination, and assessing DON-related hazards.

## 1. Introduction

Deoxynivalenol (DON; 4-deoxynivalenol; vomitoxin), a type B trichothecene and a common mycotoxin, is found in cereals [1,2]. DON increases crop scab infection and reduces the yield and quality of crops [3]. If ingested, DON causes health hazards to humans and animals even at low concentrations [4,5]. DON is related to various complications such as intestinal toxicity [6], immunotoxicity [7], neurological disorders [8], and teratogenicity [9]. DON is extremely cytotoxic, both at the cellular and molecular levels. DON first enters the cell membrane, inhibits protein, RNA, and DNA syntheses, and induces apoptosis by binding to ribosomes [10]. Acute intraperitoneal injection of lethal levels of DON into mice results in histopathological effects, including hemorrhage/necrosis of the intestinal tract, bone marrow and lymphoid tissue necrosis, kidney and heart disease [8]. These failures and dysfunction of different organs/systems can be fatal [11,12]. Many countries strictly control DON contamination in foods and feed.

Wet distiller’s grain with solubles (WDGS) is a valuable byproduct of fuel ethanol production, used as feed for livestock. Generally, WDGS is further dehydrated and dried to dried distiller’s grains with solubles (DDGS) [13]. In the United States, 83% of the feedlots use byproducts from grains, especially WDGS and DDGS in their diets [14]. In 2008, global production of bioethanol reached 76.5 hm^3^. The production rate of WDGS per liter of ethanol is in the range of 2.4–4.2 Mg/m^3^ of ethanol, indicating that WDGS is a large resource [13]. However, one study showed that ~12% of the 67 samples studied contained DON levels higher than the minimum advised level by the Food and Drug Administration, these DDGS are not only contaminated during growth but also accumulated during storage [15]. These DON-contaminated DDGS not only affect the health of humans and animals but also bring huge economic losses to farmers, and increase the global food supply burden. Therefore, a cost-effective environmentally friendly, efficient, and inexpensive detoxification method must be developed for decontaminating food crops, which would diminish the toxicity of DON-contaminated foods and reduce farmers’ losses.

Various physical, chemical, and biological treatments [16], such as photocatalytic [17], microwave treatment [18], pressure heating [19], ozone [20], and microbial feed additives [16], are used for degrading DON. However, all these strategies have various deficiencies, such as limited degradation effects, expensive equipment, unclear security, and vulnerability to environmental factors. Consequently, it is still crucial to develop an environmental, efficient, and inexpensive detoxification technology to improve the decontamination efficiency of DON.

Electrochemical oxidation (ECO) is an emerging oxidation technology [21,22,23]. During the anodization process of ECO, contaminants in the solution can be destroyed by several mechanisms. First, contaminants on the surface of the anode may be directly oxidized, and electrons are directly transferred from the contaminants to the electrodes. Second, when a high oxygen evolution electrode anode is used, contaminants may be indirectly oxidized by the active hydroxyl radicals produced at the anode. Third, when chlorine is present in the solution, a chlorine-based oxidant, such as hypochlorous acid or hypochlorite is produced. According to the direct oxidation mechanism, H_2_O molecules in the solution are discharged at the anode’s active sites M to produce adsorbed hydroxyl radicals (Equation (1)), which can potentially evolve towards the formation of chemisorbed oxygen MO (Equation (2)) [24]. Therefore, compounds containing only hydrocarbons (R) are oxidized to carbon dioxide and water (Equation (3)) or other intermediates (Equation (4)).
(1)$$M+H_{2}O\to M\left(\cdot \mathrm{OH}\right)+H^{+}+e^{-}$$
(2)$$M\left(\cdot \mathrm{OH}\right)\to \mathrm{MO}+H^{+}+e^{-}$$
(3)$$M\left(\cdot \mathrm{OH}\right)+R\to M+m\;{\mathrm{CO}}_{2}+n\;H^{+}+e^{-}$$
(4)MO+R→M+RO


If the potential is improved, more hydroxyl radicals can be produced, but it also exacerbates the anodic oxygen evolution side reaction (Equation (5)), which competes with the reaction of organics with electrogenerated electrolytic hydroxyl radicals [25].
(5)$$M\left(\cdot \mathrm{OH}\right)\to M+\frac{1}{2}O_{2}+H^{+}+e^{-}$$


In addition, when NaCl is present in the solution, the following reactions occur:
(6)$$2\;{\mathrm{Cl}}^{-}\to {\mathrm{Cl}}_{2}+2\;e^{-}$$
(7)$${\mathrm{Cl}}_{2}+H_{2}O\to \mathrm{HClO}+H^{+}+{\mathrm{Cl}}^{-}$$
(8)$$\mathrm{HClO}⇋H^{+}+{\mathrm{ClO}}^{-}$$


Compared to other oxidation degradation methods, electrochemical oxidation is advantageous due to: (i) no requirement of additional auxiliary chemicals, (ii) no secondary pollution, (iii) mild reaction conditions, (iv) easy operation, and (v) it can be combined with other treatments and fully automatized [26,27]. Thus, ECO is an environmentally friendly technology for degrading highly refractory pollutants, such as organochlorine pesticides [28,29], heavy metal [30], and insecticides [31,32] in polluted water. Research conducted by Jeon and Park [33] indicates that electrochemical technology can be combined with the alcohol fermentation industry. In addition, previous studies with ECO on microcystins [34] and ochratoxin A (OTA) [35] demonstrate the oxidative degradation potential of ECO. However, no studies have used ECO for the degradation of DON.

In this study, the ECO degradations of DON were compared using two common electrode materials, the titanium (Ti) mesh electrode and the graphite electrode. The effects of potential, initial pH, initial concentration, and recycling times on degradation were evaluated. Scanning electron microscopy (SEM) was used to observe the structural characteristics of the graphite electrode’s surface. CCK-8 and DAPI staining assays were used to evaluate the cytotoxicity of DON and its ECO products.

## 2. Results and Discussion

### 2.1. The Efficiency of Different Electrodes on DON Degradation

The degradation rate of DON was monitored using different electrodes at a constant potential of 0.6 V (Figure 1a). When ECO was performed using a Ti mesh and graphite electrodes, different results were achieved. The degradation rate of DON by both materials increased with time. After 30 min of treatment, the degradation rate of DON was 20% for the Ti mesh electrode, whereas it was more than 85% for the graphite electrode. After 60 min of electrolysis, the degradation of DON was very marginal with the Ti mesh electrode. The degradation rates of DON on the Ti mesh and the graphite electrodes were 29.7% and 97.6%, respectively. These results suggest that the graphite electrode produced better electrocatalytic performance than the Ti mesh electrode, which is in agreement with results from a previous study on wastewater treatment [36]. The significant difference was possibly obtained due to the use of the graphite electrode [37] and the formation of some intermediates [36] in our study.

The cyclic voltammograms of the Ti mesh and the graphite electrodes in a NaCl solution further illustrate the excellent performance of the graphite electrode. Figure 1b shows that the Ti mesh electrode had a lower oxygen evolution potential of 0.6 V (vs the standard Ag/AgCl). However, the graphite electrode anode had an oxygen evolution potential of 0.8 V. The higher oxygen evolution potential of the graphite electrode was translated into a better inhibition effect on the oxygen evolution reaction compared to that of Ti mesh electrode, which enhanced the formation of hydroxyl radicals. Therefore, the low cost and excellent electrocatalytic activity of the graphite electrode were ideal for the next experiments.

### 2.2. Effect of Potential on DON Degradation

The applied potential affects the electrochemical oxidation process and operational costs. Figure 2 depicts the relationship between DON degradation efficiency and potential, with 0.05 M NaCl as the supporting electrolyte. The applied potential significantly influences the electrochemical degradation efficiency. In Figure 2a, the maximum DON degradation (95.67%) was achieved at a higher applied potential of 0.6 V within 60 min. At 0.2 V, however, 53.4% degradation was achieved in 60 min. This could be attributed to the increase in current density with the applied potential. At high current density, numerous **^.^**OH and active chlorine are electrogenerated, which promotes the ECO degradation of organic compounds [38,39].

The ECO process of organic substance is closely related to the oxygen transfer reaction in a solution. On the one hand, a certain relationship exists between the electrodes’ oxygen evolution potential and the activity of electrocatalytic oxidation of organics [40], because these oxygen-active groups participate in the oxidation of organics. On the other hand, since the anodic oxygen evolution reaction competes with the reaction of organics and electrogenerated electrolytic hydroxyl radicals, a low oxygen evolution potential would reduce the electrochemical oxidation efficiency [41]. Although an increase in the potential and time would benefit the electrochemical oxidation of organic compounds, it would also increase the generation of oxygen and hydrogen, causing unnecessary power consumption. In Figure 2b, the degradation rate of DON was 86.4% after 30 min of treatment at a constant potential of 0.5 V. Prolonging treatment time beyond 30 min did not affect the degradation rate significantly. A potential of 0.5 V and treatment time of 30 min was selected in the following experiment to minimize the adverse reaction of oxygen evolution caused by the increased potential.

To further study the ECO reaction process and optimize the reaction conditions, the kinetic order of electrochemical oxidation was determined at the potential of 0.5 V. As the potential was constant during the reaction and the concentration of DON changed with time, the reaction was assumed to be a first-order reaction (dC/dt = k C) [42]. Following the logarithmically transformed equation (Equation (9)), the $\ln\left(C/C_{0}\right)-t$ curve is shown in Figure 3. A straight line with K = 0.0575 was obtained. The curve exhibited a good linear relationship (R^2^ = 0.991), suggesting that the electrochemical oxidation reaction was a first-order kinetic reaction.
(9)$$\mathrm{lnC}/{\mathrm{lnC}}_{0}=-k\;t$$
C = concentration of DON at time t (mg/L);C_0_ = initial concentration of DON (mg/L);k = thermal degradation rate constant (min^−1^);t = processing time (min).


### 2.3. Effect of Initial Concentration on DON Degradation

As a first-order reaction, the reaction rate depends on the concentration of the reactants. The effects of different initial concentrations on the degradation efficiency were investigated. In Figure 4, the degradation rates of DON were 79.8, 86.3, 90.5, 93.2, and 93.9% at the initial concentrations of 2, 4, 6, 8, and 10 mg/L, respectively. Therefore, the degradation efficiency of DON increased with increasing concentration.

According to the national standard (GB 2761–2011) in China, the maximum allowable content of DON in grains and their finished products is generally 1000 μg/kg. Figure 4 shows that DON solutions with initial concentrations of 2, 4, 6, 8, and 10 mg/L were reduced to 0.40, 0.57, 0.57, 0.53, and 0.60 mg/L, respectively, after 30 min of electrochemical oxidation. Therefore, all concentrations of DON in this study could be degraded to an acceptable range by ECO. In summary, when this method was applied to practical use, the concentration of DON within a certain range in the environment could be degraded to an acceptable level according to the national standard of China. The concentration of DON had an insignificant influence on the degradation effect.

### 2.4. Effect of Initial pH on DON Degradation

An important operating parameter in the electrochemical oxidation of organic compounds is the pH value, since in most cases the removal efficiency is maximal at an optimum pH value [43]. The effect of the initial pH value on the ECO process was monitored at the initial pH of 4.0, 5.0, 6.0, 7.0, and 8.0 at the applied potential of 0.5 V. Figure 5 shows that the DON degradation rate decreased with an increase in initial pH. The maximum DON degradation rate reached 86.2% and 86.4% at initial pH 5.0 and 4.0, respectively, while the DON degradation rate was observed to be 71.1% at the initial pH 8.0.

Electrogeneration of many oxidizing species depends upon the nature of the electrolyte and pH [44]. NaCl was selected as the electrolyte in this experiment, and therefore, many active chlorine species, such as Cl_2_, ClO^−^, and HClO were generated in the solution. In the process of electrolysis, until pH 3.0, the main active chlorine species was Cl_2_ (aq), while the main substance in the pH range of 3–8 was HCLO and the main substance for pH > 8 was ClO^−^ [45]. In addition, Tang [46] reported that high acidity could enhance the formation of free radicals, therefore DON was more susceptible to oxidation in this study. DON will lose certain oxidizing ability in basic media and the oxidation proceeds slowly [47]. The experimental results in this article were consistent with the above conclusions, and the maximum degradation rate was achieved at the pH 4.0 and 5.0. pH 5.0 was selected as a condition for the following experiments based on cost considerations.

### 2.5. Electrolytic Stability and SEM of the Graphite Anode

The electrocatalytic stability of graphite was studied under optimal conditions. In Figure 6, the ECO degradation rate of DON by the graphite electrode after three cycles is shown. The ECO degradation efficiencies of graphite electrodes for DON were 87.8, 84.7, and 81.1% at the first, second, and third recycling, respectively. After each treatment, DON concentrations could reach below the acceptable value according to the national standard (GB 2761–2011). The results confirm that DON can be catalytically degraded by the graphite electrode, and the method shows good reproducibility and electrolytic stability. The graphite electrode displayed excellent performance mainly due to two reasons: firstly, the surface properties of the graphite electrode allowed the adsorption of DON on its surface; secondly, the graphite electrode acted as an electron-transfer catalyst thanks to its excellent electrical conductivity [48,49].

The ideal crystal structure of graphite is widely studied [50]. Graphite has higher porosity compared to other carbon materials, such as glassy carbon and carbon black [40]. The material’s morphology can be investigated by SEM [47]. SEM images of the graphite anode before (fresh) and after three cycles of electrochemical treatment of DON are shown in Figure 7. Figure 7a indicates a porous structure of the fresh graphite surface, suggesting that the graphite electrode’s surface provides many active sites for the attachment of DON, which would further increase the rate of electrochemical degradation. Figure 7b shows that the porous structure on the surface of the graphite electrode were similar to those shown in Figure 7a, but the pore size was slightly reduced, possibly due to the formation of various oxidants during the electrocatalytic process. In addition, graphite shows high affinity for certain substances, including aromatic compounds, and its adsorption kinetics is almost 40 times that of activated carbon [51]. The target compounds’ rate of adsorption at the surface of the electrode strongly affects the electrochemical degradation rate of the target compounds [52]. Some special affinity between the graphite electrode and DON in the electric field might significantly influence the degradation of DON. Therefore, the excellent electrical conductivity, porous structure, and special affinity of the graphite electrode surface accelerated the degradation of DON.

### 2.6. Cytotoxicity Assay of DON and Its Electrochemically Oxidized Products

DON produces various toxic effects in the cells of livestock and humans. Once ingested, DON harms the epithelial cells of the gastrointestinal tract. Normally, the epithelial cells of the gastrointestinal tract serve as the first line of defense [53]. DON significantly inhibits cell proliferation in vitro due to its cytotoxic property [54]. DON is a potent inhibitor of RNA, DNA, and protein synthesis at the cellular level. First, DON enters the cell membrane, binds to the 60S subunit of the eukaryotic ribosome, and then disrupts the activity of the peptidyl transferase. This behavior induces a signal-mediated ribotoxic stress response, which inhibits cell proliferation. In addition, DON can also be converted into different intermediates, which are more polar and less cytotoxic than the parent compound [55].

The cytotoxicity of DON and its electrochemically oxidized products was assessed using the CCK-8 assay (Figure 8). Figure 8a shows DON reduced cell viability in a dose- and time-dependent manner. The inhibition rates were 32.0, 39.4, 48.3, 55.7, and 60.7% at DON concentration of 2, 4, 6, 8, and 10 mg/L, respectively, after 24 hours of culture. Meanwhile, the inhibition rate of GES-1 cells was 32.0, 60.5, and 81.5% after incubating for 24, 48, and 72 h, respectively, at 2 mg/L DON. The inhibition rate reached 80% at all concentrations after 72 hours of DON treatment. Similar experimental phenomena have also been confirmed in the study by Yang et al. [56]. In addition, Dai’s [57] research also demonstrated that DON not only causes dose-dependent cytotoxicity, but also causes time-dependent cytotoxicity of mouse endometrial stromal cells. According to Figure 8b, the viability of GES-1 cells was greatly reduced after incubation at 2 mg/L compared to the control group. However, the cell inhibition rate of electrochemically oxidized products was significantly reduced compared to the DON-treated group. After 72 hours of culture, the cell inhibition rate at 2 mg/L of DON was 81.5%, and the cell inhibition rate of the electrochemically oxidized products was 15.86%. 

A DAPI staining assay was conducted to observe the changes in quantity and morphologies of GES-1 cells (Figure 9). Fluorescence microscopy (50 µm) showed that the number of cells was significantly reduced after DON treatment compared to the control group. This was caused by the fact that during the culture, the cells affected by DON retracted from their adjacent cells, rounded up, and eventually floated into the medium, suggestive of apoptosis [58]. Dramatic proapoptotic morphological changes were observed in GES-1 cells (10 um) compared to the blank control, including cell shrinkage, chromatin condensation, and apoptotic body formation (arrows) [59]. In contrast, the normal cell nuclei of the control group emitted a uniform pale blue fluorescence. Interestingly, no significant changes were observed in ECO-treated cells compared to the control group. 

Finally, these results show that the cytotoxicity of the degraded products was significantly diminished after electrolysis because DON was degraded to certain nontoxic or less-toxic intermediates after electrochemical oxidation.

### 2.7. Electrochemical Oxidation of DON in WDGS

Figure 10a shows the degradation rate of DON in WDGS at a constant potential of 0.5 V. After 30 min of ECO treatment, the degree of DON degradation rate was 61.48% from the initial concentration of DON at 23.2 mg/Kg. Then the degradation rate reached 86.37% at 60 min. In Figure 10b, the HPLC chromatogram clearly shows the degradation of DON in DDGS. The peak at 30 min was significantly diminished compared to that at 0 min, and the peak of DON almost disappeared at 60 min (DON concentration < 0.3 mg/L). No new peaks were observed, indicating that there may be no intermediate product formation during the ECO process. 

The color of DDGS was highly correlated with the nutritional properties of DDGS [60] and consumer acceptance. The color parameters (L*, a*, and b*) of the DDGS are listed in Table 1. L* denotes the measurement of brightness, a* represents the red-green coordinates, and b* measures the blue-yellow coordinates of the product [61]. After ECO treatment, the a* and b* values of DDGS slightly increased, thus the color was deepened, but these changes were not statistically significant. In addition, previous studies have shown that as the brightness and yellowness of DDGS reach a certain threshold (L* between 28 to 34, and b* between 15 to 20), the apparent and true digestibility of amino acids declined [62]. Therefore, the effect of ECO treatment on the color of DDGS was not significant and may not have affected the apparent and true digestibility of amino acids. In summary, ECO exerted a good degradation effect on DON in WDGS, and the commercial value of DDGS did not decrease significantly after ECO treatment.

## 3. Materials and Methods

### 3.1. Materials

The graphite electrode and the Ti mesh electrode were purchased from Xuri Metal Materials Sales Co. Ltd., Tianjin, China. The human gastric epithelial cell line GES-1 was stored in the Key Laboratory of Biological Resource and Ecological Environment of Chinese Education Ministry. The DON standard (analytical standard, purity ≥ 99%; CAS: 51481-10-8) was purchased from Immunos Singapore. The cell counting kit-8 (CCK-8) was purchased from Meilun Biotechnology, China; Dulbecco’s modified Eagle’s medium (DMEM) was purchased from HyClone, USA; fetal bovine serum (FBS) was purchased from Gibco, USA; and 4,6-diamidino-2-phenylindoledihydrochloride (DAPI) was purchased from Solarbio, China. HPLC grade Acetonitrile and methanol were used in this study, and purchased from Sigma, USA. WDGS was sampled at the winery of Yinshan Hongzhan Industrial Co., Ltd., Ziyang City, Sichuan Province, China, and sealed at room temperature. All other chemicals and solvents were of analytical grade and obtained from Shudu Co. Ltd., Chengdu, China.

### 3.2. Experimental Setup

A two-chamber electrolytic cell was used in this experiment. The volume of each cell was 15 cm^3^ (the volume of the cells were 100 cm^3^ when the WDGS was treated). The anode and the cathode cells were separated by a cation exchange membrane. A 50 mm gap was maintained between the cathode and anode. The CHI760E electrochemical workstation (Chenhua, China) was used as a power supply for the reactor. Samples were magnetically stirred at a speed of 1000 r/min for thorough mixing. The anode cell was filled with the DON standard solution or WDGS and the cathode cell was filled with 0.02 M NaCl solution as a supporting electrolyte. All the experiments were performed at room temperature.

### 3.3. The Efficiency of Different Electrode Materials on the Degradation Rate of DON 

Electrode materials directly affect ECO efficiency. According to the previous experiments shown in Appendix A and Appendix A, two different electrode materials, the Ti mesh electrodes and the graphite electrodes, were compared under a constant potential of 0.6 V. 

The electrocatalytic properties of the two materials were compared through cyclic voltammetry (CV) using a three-electrode system. The working electrode was either a Ti mesh electrode or a graphite electrode, the auxiliary electrode was a platinum wire electrode, the reference electrode was an Ag/AgCl electrode. The potential output was controlled by Chenhua station CHI760E, the scanning speed was maintained at 100 mV/s, and an electrolyte solution of 0.02 M NaCl aqueous solution was used.

DON was quantified using a HPLC system (1525, Waters, USA) equipped with a UV/Vis detector and a Waters-C18 chromatographic column (250 mm × 4.6 mm, 5 um; Waters, USA). Conditions for HPLC analyses were: mobile phase, acetonitrile/water (12:88, *v*/*v*); flow, 1.0 ml/min; temperature, 30 °C; detection wavelength, 218 nm; injection volume, 20 μL; and retention time, 10 min. 

### 3.4. Electroche of DON with the Graphite Electrode

The graphite electrode was used in this experiment. DON solutions were electrochemically oxidized under constant potentials of 0.2, 0.3, 0.4, 0.5, and 0.6 V. Samples were collected every 10 min during the experiment, and DON concentrations were analyzed by HPLC. The effects of different initial concentrations and initial pH on the degradation of DON were determined. A series of solutions with the initial pH of 4 to 8 was prepared using a NaOH solution of 3 M or a HCl solution of 2 M, as necessary. The initial DON concentrations varied from 2 to 10 mg/L and adjusted by a 0.2 M NaCl solution.

### 3.5. Electrolytic Stability of the Graphite Electrode

The reusability and long-term performance are key factors for developing economically viable DON degradation processes. We performed the ECO experiments over three cycles under optimal condition. The next experiment was performed without treating the graphite electrode after each experiment. 

### 3.6. Cytotoxicity Assay of DON and Its Electrochemically Oxidized Products

The inhibition effect of DON on human gastric epithelial cells (GES-1) was assessed using the CCK-8 assay. Cells were divided into DON, ECO, and control groups. ECO products were obtained by electrocatalyzing a 2 mg/L solution of DON for 30 min at 0.5 V. GES-1 cells in the logarithmic growth phase were maintained in DMEM media, supplemented with 10% FBS and antibiotics (100 U/mL of penicillin and 100 mg/L of streptomycin), and incubated in 96-well plates (2 × 10^3^ cells/well). After culturing for 24 h at 37 °C with 5% CO_2_, cells were incubated with DON or ECO products, an equal amount of water was added as a control. After 24, 48, and 72 h cultures, the medium was replaced with DMEM containing 10% CCK-8. The cells were further incubated for 2 h at 37 °C, then the absorbance of each sample was measured at 450 nm using a microplate reader to determine the inhibition effect of DON and its ECO products on GES-1 cells.
In=(CO−EO)/CO×100%
In: Inhibition ratio;EO: Optical density(OD) value of the experimental groups;CO: OD value of the blank groups.


### 3.7. DAPI Staining

DAPI staining was used to visualize nuclear changes and apoptotic body formation. GES-1 cells were seeded on 6-well plates (1 × 10^5^ cells/well). According to the operation in Section 3.6, cells were divided into DON, ECO, and control groups. After culturing for 24 h, the attached cells were fixed in situ using 75% alcohol overnight at 4 °C. The cells were then treated with 0.25 mg/mL DAPI solution, and incubated in the dark for 15 min at room temperature. The stained cells were observed under a fluorescence microscope (Leica DMi 8, Germany).

### 3.8. Electrochemical Oxidation of WDGS

According to the experimental results in Section 3.4, WDGS with a known DON concentration was electrocatalyzed at a potential of 0.5 V, samples were collected every 10 min during the experiment, and DON concentrations were analyzed using HPLC. The pH of WDGS used in this experiment was ~4.3, which was slightly higher than the pH of the samples studied by Lyberg et al. [63], but it was still favorable for electrocatalysis. Therefore, the pH of WDGS was not adjusted in this experiment. DON was extracted using an immunoaffinity column (DON IAC; Copure, China), and the extraction method followed the modified procedure of the China National Standard (GB/T23503-2009). DDGS was obtained after drying WDGS at 100 °C for 30 min. Then, 10 g of powder was added to 100 mL of water and shaken for 30 min, then filtered using filter paper, and 1 mL of supernatant was transferred to the column. Finally, DON was eluted with 1 mL of HPLC grade methanol. The color changes in DDGS during ECO were examined using a colorimeter (KONICA MINOLTA, Japan) and expressed as L*, a*, and b*. 

### 3.9. Statistical Analysis

Three parallel experiments were performed for each set of treatments, and the data from the independent replicates were combined and calculated using mean ± SD. The Shapiro–Wilk test was used to test the normality of the data. Normally distributed data (*p* > 0.05) were evaluated by one-way ANOVA with multiple comparisons according to Tukey’s test (*p* < 0.05). Data that were not normally distributed (*p* < 0.05) were analyzed by the nonparametric Kruskal–Wallis test (*p* < 0.05).

## 4. Conclusions

The degradation of DON and its contaminated WDGS by ECO with graphite as an electrode was studied. The DON content was greatly reduced in the treated DDGS, and the ECO treatment did not affect the appearance of DDGS. High potential was the primary factor causing DON degradation, and acidic conditions also contributed to the degradation of DON. It was due to both direct and indirect oxidation existing during ECO, and the acidic condition was favorable for the formation of strong oxidizing intermediates. The CCK-8 assay and DAPI staining process demonstrated the inhibitory effect of DON against the proliferation of GES-1 in a dose- and time-dependent manner. Furthermore, the toxicity and hazards were greatly attenuated after ECO treatment. ECO is a fast, safe, and effective method for reducing mycotoxins in foods. This work provides an idea for further detoxification of DON contaminated grains and foods. Further experiments and investigations are required to determine the actual economic benefits of implementing ECO technology as part of the integrated management program.

## Figures and Tables

**Figure 1 toxins-11-00478-f001:**
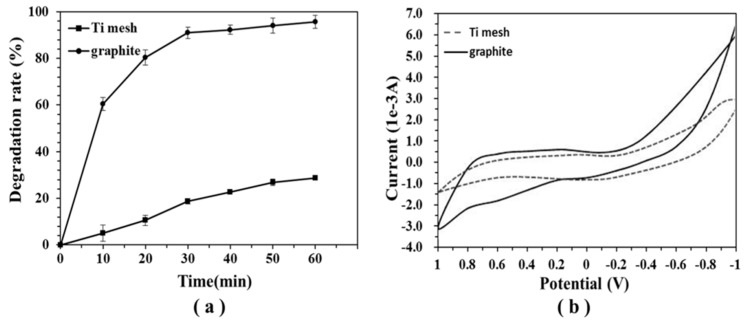
Effect of the electrode materials. (**a**) Degradation rate when using different electrodes (initial deoxynivalenol (DON) concentration of 4 mg/L, supporting electrolyte concentration of 0.02 M of NaCl aqueous solution, applied potential of 0.6 V). The values were expressed as means ± standard deviation (SD) (n = 3). (**b**) The cyclic voltammograms of the Ti mesh and the graphite (scan rate of 100 mV/s, supporting electrolyte concentration of 0.02 M of NaCl aqueous solution).

**Figure 2 toxins-11-00478-f002:**
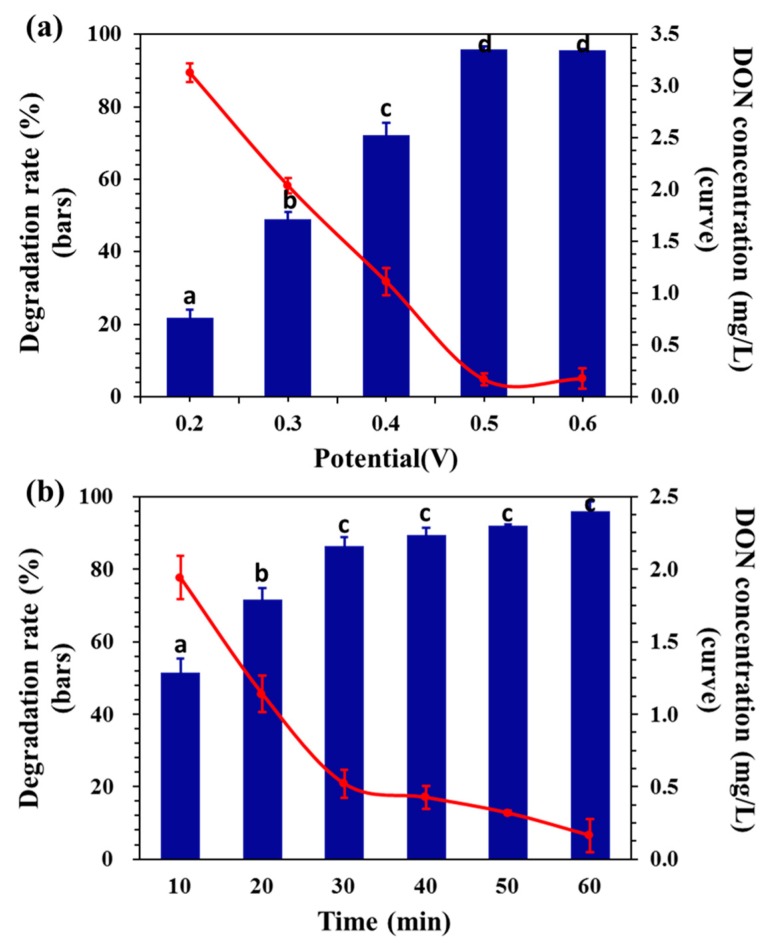
(**a**) Degradation rate at different potentials after 60 min of treatment. (**b**) Degradation rate at different points of time at 0.5 V (initial DON concentration of 4 mg/L, supporting electrolyte concentration of 0.02 M of NaCl aqueous solution). Means with unlike letters (**a**–**c**) differed significantly as per the Kruskal–Wallis test (*p* < 0.05), and the values were expressed as means ± SD (n = 3).

**Figure 3 toxins-11-00478-f003:**
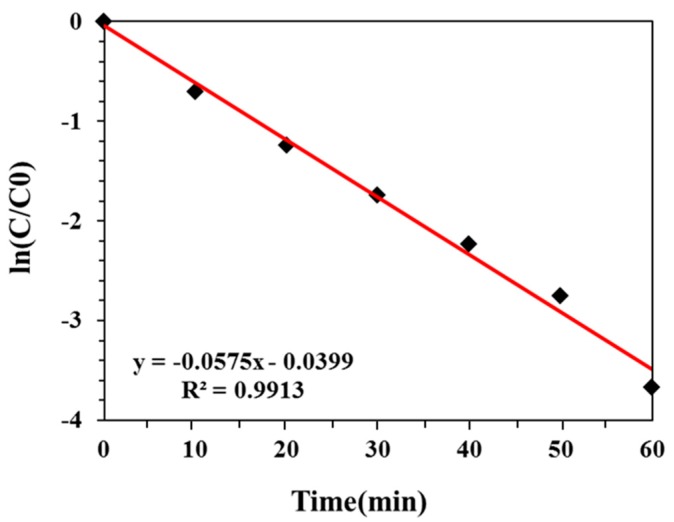
Variation of ln(C/C_0_) with time at room temperature (initial pH of 5.0, initial DON concentration of 4 mg/L, supporting electrolyte concentration of 0.02 M of NaCl aqueous solution).

**Figure 4 toxins-11-00478-f004:**
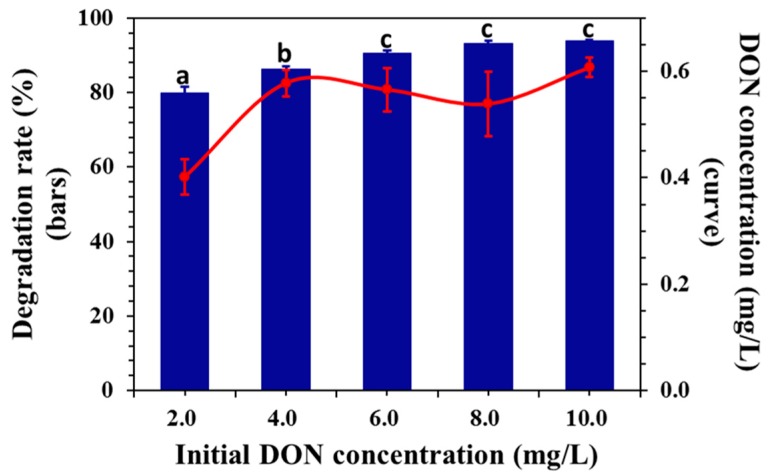
Effect of the initial concentration (potential of 0.5V, initial pH of 5.0, supporting electrolyte concentration of 0.02 M of NaCl aqueous solution). Means with unlike letters (**a**–**c**) differed significantly by the Kruskal–Wallis test (*p* < 0.05), and the values were expressed as means ± SD (n = 3).

**Figure 5 toxins-11-00478-f005:**
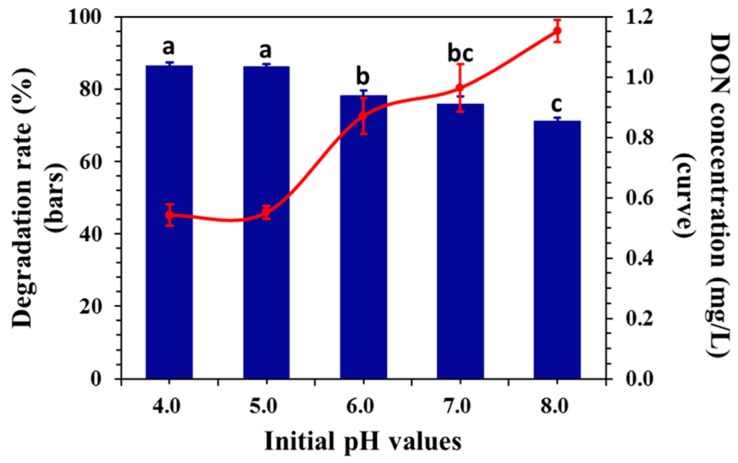
Effect of the initial pH (potential of 0.5 V, initial DON concentration of 4 mg/L, supporting electrolyte concentration of 0.02 M of NaCl aqueous solution, treatment time of 30 min). Means with unlike letters (**a**–**c**) differed significantly according to Tukey’s test (*p* < 0.05), and the values were expressed as means ± SD (n = 3).

**Figure 6 toxins-11-00478-f006:**
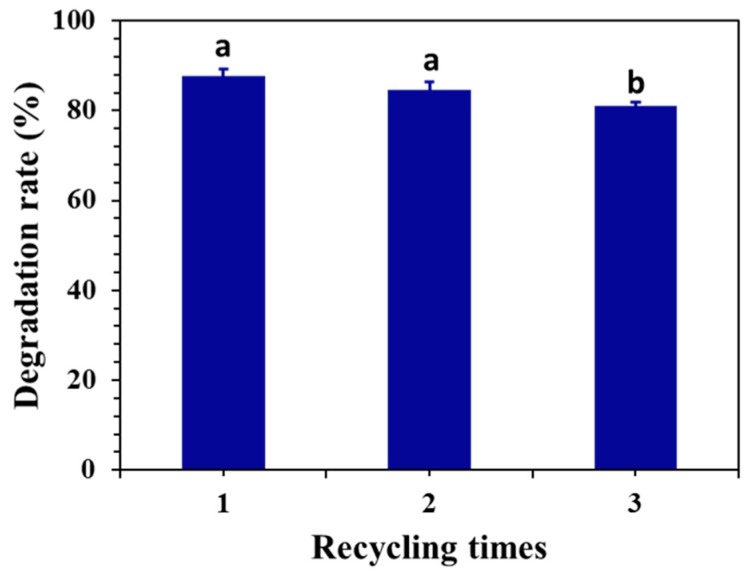
Effect of the recycling time on the degradation of DON. The experiment was performed without using the graphite electrode after each experiment (applied potential is 0.5 V, initial pH is 5.0, initial DON concentration is 4 mg/L, supporting electrolyte concentration is 0.02 M of NaCl aqueous solution, treatment time of 30 min). Means with unlike letters (**a**,**b**) differed significantly according to Tukey’s test (*p* < 0.05), and the values were expressed as means ± SD (n = 3).

**Figure 7 toxins-11-00478-f007:**
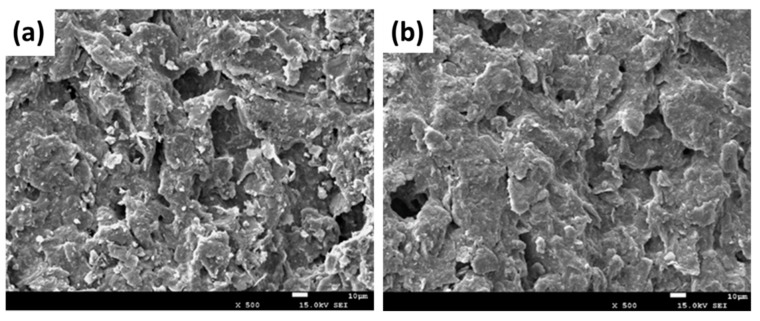
**Scanning electron microscopy** (SEM) images of the graphite anode (**a**) before treatment and (**b**) after treatment. The pore size was reduced slightly after treatment (initial DON concentration is 4 mg/L, initial pH is 5.0, treatment time is 30 min, recycled three times).

**Figure 8 toxins-11-00478-f008:**
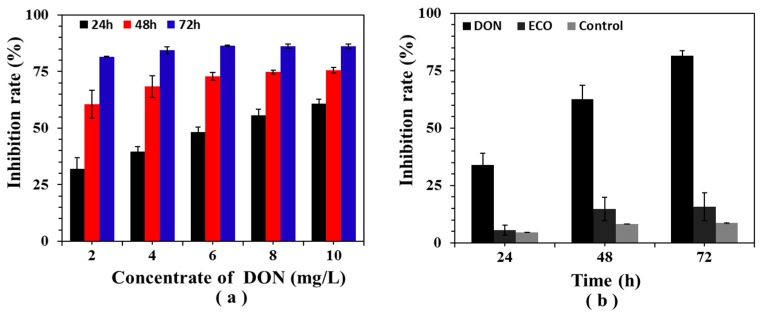
The inhibition rate of DON and its electrochemically oxidized products. (**a**) The cells were treated with different concentrations DON for 24, 48, and 72 h. (**b**) The cells were treated with 2 mg/L of DON and its electrochemically oxidized products (ECO) for 24, 48, and 72 h, an equal amount of water was added as a control. The values were expressed as means ± SD (n = 3)

**Figure 9 toxins-11-00478-f009:**
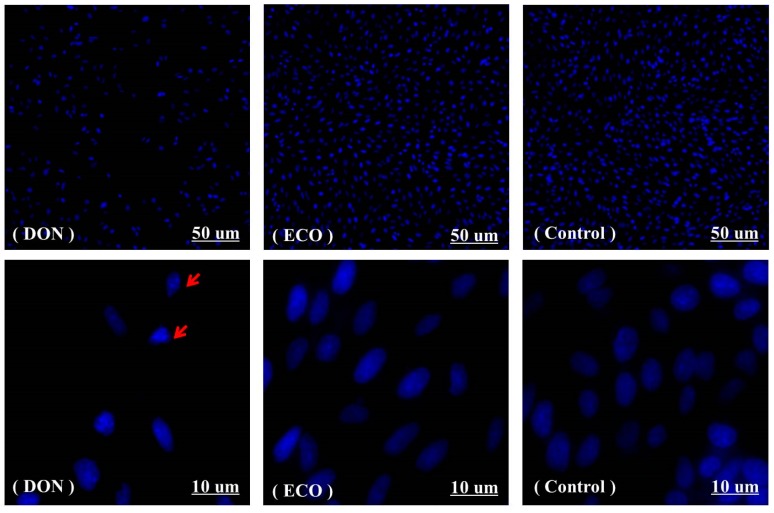
The changes in quantity and morphology of GES-1 cells by DAPI staining. The cells were treated with 2 mg/mL of DON (DON) and its electrochemically oxidized products (ECO), with an equal amount of water as a control (Control) for 24 h. The DAPI dye stains the morphologically normal nuclei to uniform pale blue, whereas DON-treated cells demonstrated smaller nuclei, stained with brilliant blue (arrows). The images are representative of three independent experiments.

**Figure 10 toxins-11-00478-f010:**
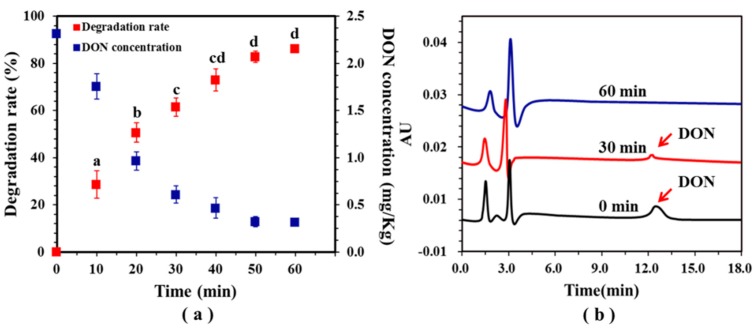
Degradation of DON in wet distiller’s grain with solubles (WDGS) byelectrochemical oxidation (ECO). (**a**) Changes in the degradation rate and concentration of DON in WDGS. Means with unlike letters (**a**–**c**) differed significantly according to Tukey’s test (*p* < 0.05), and the values are expressed as means ± SD (n = 3). (**b**) HPLC chromatograms of DON at different times. The supporting electrolyte is NaCl aqueous solution with concentration of 0.02 M, and applied potential is 0.6 V.

**Table 1 toxins-11-00478-t001:** Color changes of electrochemical oxidation (ECO)-treated dried distiller’s grains with solubles (DDGS).

Treatment	L*	a*	b*
Before ECO	62.72 ± 0.09 ^a^	8.97 ± 0.43 ^a^	35.92 ± 1.54 ^a^
After ECO	62.59 ± 0.66 ^a^	9.78 ± 0.94 ^a^	36.79 ± 0.88 ^a^

Values are presented as means ± SD (n = 3). Values with the letter “a” indicate no significant difference (*p* > 0.05).

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
