# Peer review of "The Degradation of Deoxynivalenol by Using Electrochemical Oxidation with Graphite Electrodes and the Toxicity Assessment of Degradation Products"

_toxins, 2019, doi:10.3390/toxins11080478_

Round 1

Reviewer 1 Report

Review Report

A brief summary

The study was aimed to demonstrate the effectiveness of electrochemical oxidation in degrading of deoxynivalenol (DON). In addition, using human gastric epithelial cells (GES-1) in an in vitro toxicity assay, the authors showed that the degradation products were non-toxic or less-toxic.

Broad comments

The experiments are logic, seemed to be carefully done and the paper is thoroughly written. Each assay was properly described, and the results were systematically presented and discussed.  

Specific comments

General comment about the introduction:

To clarify the motivation for the work, I would suggest mentioning that in which points of crop production one can use this degradation technique (pre-harvest, harvest or post-harvest) and how an electrochemical oxidation which is performed in a solution media can practically/commercially be used in preventing DON in feed/food chain as solid media.  Please use the proper citation to support your idea.

This is also of interest to cite previous studies on ECO degradation of other mycotoxins. One example is reducing ochratoxin A (OTA) by ECO. Add relevant citations.  

Page 1 line 7: Please replace toxins with mycotoxins.

Page 1 lines 15 and 16: CCK-8, DAPI and GES-1 should be defined since it is used for the first time in the manuscript.

Page 1 line 31: In the case of teratogenic effects of DON, the results from previous studies vary from multiple teratogenic effects to no teratogenic effects. I would suggest that DON-induced effects can be mentioned in the following order (add some other important effects as well):  

-          Replace also immunosuppression with immunotoxicity since DON can be both immunosuppressive and immunostimulant.  

Suggested order: Intestinal toxicity, immunotoxicity, neurological disorders and teratogenicity.

Page 1 lines 34 and 35: The death can occur due to multiorgan dysfunction. Failure and dysfunction of different organs/systems such as gastrointestinal and immune system can be associated with death. It is also worth mentioning that the acute exposure with high doses of DON can be fatal. Reformulate this sentence.     

Page 1 line 40:  Reference number 9 does not correspond to the information given in this sentence. Please provide another reference or omit it.

Page 1 line 40: Cite a more comprehensive study that reviewed chemical, physical and biological treatments. One example is a review by Awad et al [1]

Page 1 line 41: Reference number 12 covers ozone treatment but not microwave and pressure heating. Please provide proper citations for these two treatments as well.

Page 1 line 41: Chlorine dioxide in the cited study was used to control Fusarium graminearum and not to degrade DON. Reformulate or omit the cited information.

Page 2 line 69: In the (Eq.7) Cl à Cl–

Page 2 line 90: degradation effect à degradation rate

onàby

Page 3 line 97: Add some content and explain more about previous findings. In the cited study, the effects of different electrodes on the treatment of wastewater were investigated. Please clarify this to prevent misunderstanding.

Page 3 Figure 1: It is a general comment about the following figures (Figures 1, 3, 5, 6, 7 and 9).

Explain the type of error bars in the figure captures. Clarify whether they are SD or SEM…

Page 7 line 205: Electrolysis à Electrolytic

Page 7 line 212: Electrolysis à Electrolytic

Page 9 line 254: ribose toxic à ribotoxic

Page 9 line 254: leads à lead

Page 9 line 266: Relevant reference should be added to discuss/or confirm this finding. It has been shown in the previous study that cell viability decreased, and that cell death rate increased in a dose and time-dependent manner [2]

Page 11 line 318: Provide the name of the companies that electrodes were obtained.

Page 12 line 328: cm3à cm3

Page 12 line 367: groupà groups

Page 13 line 392: It is important to mention which type of ANOVA has been used.

How did you test the normality of the data?

Did you use repeated measured ANOVA for the repeated measured parameters based on the time points (data presented in figures 1 and 3)? If not, why?

Please provide more information on the statistical analysis performed in different data types.

References:

1.            Awad, W.A.; Ghareeb, K.; Bohm, J.; Zentek, J. Decontamination and detoxification strategies for the Fusarium mycotoxin deoxynivalenol in animal feed and the effectiveness of microbial biodegradation. Food Addit. Contam., Part A 2010, 27, 510–520, doi:10.1080/19440040903571747.

2.            Yang, Y.; Yu, S.; Liu, N.; Xu, H.; Gong, Y.; Wu, Y.; Wang, P.; Su, X.; Liao, Y.; De Saeger, S., et al. Transcription Factor FOXO3a Is a Negative Regulator of Cytotoxicity of Fusarium mycotoxin in GES-1 Cells. Toxicological sciences : an official journal of the Society of Toxicology 2018, 166, 370-381, doi:10.1093/toxsci/kfy216.

Reviewer 2 Report

The manuscript "The degradation of deoxynivalenol by using electrochemical oxidation with graphite electrodes and the toxicity assessment of degradation products" reports novel results about the possible degradation of the mycotoxin by electrochemical oxidation. Furthermore it assess the cytoxicity of the possible degradation products. 

The results are interesting but the extrapolation to real situations is not addressed. My suggestion before the manuscript can be published is to verify the suitability of the developed protocol to eliminate the levels of DON in food matrices. In addition, the safety for consumers of the use of this technique in food should also be adressed.

In addition, the manuscript should be revised by a native english speaker.

In the conclusions of the summary, it is state that findings provide new insights into the reduction of grain and food contamination, however this has not been explored in this study.

Finally, Figure 5 is not clear respect to the description in the text.
